

# Exploring the relationships between pre-pregnancy BMI, gestational weight gain, and nutritional intake: a real-world investigation in Shandong, China

Juan Zhang, Xue Wang, Ping Zhu, Xiaoge Huang, Xingru Cao and Junmin Li

Jinan Maternity and Child Care Hospital, Jinan, China

## ABSTRACT

This study investigated the associations between gestational weight gain (GWG), pre-pregnancy body mass index (BMI), and prenatal diet quality in pregnant women from Shandong, China. We analyzed a sample of 532 early-stage pregnant women registered at an outpatient clinic. Diet quality was evaluated using the Chinese Healthy Dietary Index for Pregnancy (CHDI-P), encompassing three dimensions: diversity, adequacy, and limitation, with an overall score out of 100. Dietary intake was documented *via* 24-h dietary recalls spanning three consecutive days and subsequently translated to a CHDI-P score. At the time of enrollment, BMI was measured on-site and classified as underweight (<18.5), normal weight (18.5–24.9), overweight (25.0–29.9), and obese (≥30.0). Pregnant women were also categorized into inadequate, adequate, and excessive weight gain groups based on their GWG. We employed a Tukey-adjusted generalized linear model to compare the CHDI-P scores between the pre-pregnancy BMI groups and GWG groups. The results revealed that the underweight group had significantly higher total scores and limitation total scores on the CHDI-P ($p < 0.001$). Conversely, the overweight and obese groups were more susceptible to suboptimal dietary quality. Notably, the inadequate weight gain group displayed significantly elevated food adequacy scores compared to the other two groups ($p < 0.05$). This indicates that greater GWGs do not necessarily align with principles of adequate nutrition.

## INTRODUCTION

The association between maternal and newborn health is profound and there are ongoing efforts to improve postpartum outcomes (*Moller et al., 2019*). The nutritional status of expectant mothers throughout gestation is a subject of interest due to its significant influence on maternal health and pregnancy outcomes (*de Freitas et al., 2022*; *Henriksen, 2006*; *Teede et al., 2022*). Excessive nutrition or gestational overweight elevates the risk of fetal anomalies and augments the probability of diabetes in both mother and neonate (*Henriksen, 2006*). Frequent consumption of meat, sugared beverages, and sweetened snacks heightens the risk of pre-eclampsia (*Brantsæter et al., 2009*), while a diet rich in fats

Corresponding authors
Xingru Cao, 1399556277@qq.com
Junmin Li, jnfyljm@163.com

and sugars correlates with an increased likelihood of preterm labor (*Grieger, Grzeskowiak & Clifton, 2014*). In economically underdeveloped regions, maternal underweight is common and is primarily attributed to the insufficient caloric intake of expectant mothers. This deficiency heightens the risk of stillbirths, neonatal fatalities, and low birth weight (LBW) in neonates (*Patel et al., 2018*).

Prior research has examined the prenatal dietary quality of expectant mothers. In the United States, the Healthy Eating Index-2010 score for these women stands at a mere 50.7 out of 100 (*Shin, Lee & Song, 2016*). Notably, a majority of these expectant mothers fall short in consuming adequate fiber, grains, fruits, and vegetables, yet their sodium and fat consumption surpassed recommended levels (*Laraia, Bodnar & Siega-Riz, 2007*; *Rifas-Shiman et al., 2009*). This dietary imbalance correlates with the 22% obesity rate observed among pregnant women (*Kim et al., 2007*). A study examining a cohort of expectant mothers in Canada revealed analogous findings. Furthermore, it identified that women with lower educational backgrounds residing in urban areas exhibited an elevated risk of suboptimal dietary quality (*Savard et al., 2019*). In China, the predominant dietary concerns among expectant mothers across various regions include imbalances and insufficient intake of numerous micronutrients (*Dong & Yin, 2018*). Additionally, the prevalence of overweight and obesity has surpassed 20% of this demographic (*Teede et al., 2022*), presenting a significant challenge. Consequently, examining the prenatal nutritional and dietary patterns of expectant mothers and devising a well-balanced dietary plan is crucial. Regrettably, most research data in this domain originates from developed Western nations. There are limited studies focusing on the Chinese population and those that have are primarily restricted to major cities such as Beijing (*Sun et al., 2020*). China's regional developmental disparities warrant attention. Empirical research from mid-sized cities and rural regions can offer a holistic insight into the dietary challenges confronting expectant mothers in the country.

A widely acknowledged measure of an expectant mother's nutritional status is the pre-pregnancy body mass index (BMI) (*Laraia, Bodnar & Siega-Riz, 2007*; *Uno et al., 2016*). Empirical research has established that both elevated and reduced BMI values influence pregnancy outcomes (*Aji et al., 2022*; *Tang et al., 2021*; *Vats et al., 2021*). Nonetheless, this metric has limitations, particularly its presumption of consistent dietary habits throughout gestation—a premise that is challenging to uphold (*Forbes et al., 2018*). Consequently, some scholars suggest that gestational weight gain (GWG) may more dynamically reflect shifts in dietary quality, either as an alternative or in tandem with prenatal BMI, and may be used to assess the nutritional health of pregnant individuals (*Cano-Ibáñez et al., 2020*; *Guelinckx et al., 2008*). The objective of this research was to investigate the associations between both pre-pregnancy BMI and GWG with prenatal dietary quality within a mid-sized urban cohort. This exploration sought to identify a subgroup particularly susceptible to compromised dietary quality. We posited that expectant mothers with both pre-pregnancy BMI and GWG within standard parameters would exhibit superior prenatal dietary quality.

## METHODS

### Participants

The participants in this study were expectant mothers who visited the obstetrics outpatient clinic at Jinan Maternal and Child Health Hospital in Shandong Province between January 2021 and December 2022. Participants had to meet the following criteria: (1) Chinese citizenship; (2) pregnancy ≤12 gestational weeks; (3) have an established maternity record at the institution and have undergone routine obstetric check-ups; (4) face-to-face completion of the survey; (5) plan to give birth at the hospital; and (6) provide signed informed consent. Exclusion criteria included: (1) non-local migrant populations unable to attend routine check-ups; (2) individuals with metabolic disorders or chronic conditions such as tumors or tuberculosis.

The study protocol received approval from the Jinan Maternal and Child Ethics Committee Health Hospital (No. 2023-1-029). All participating individuals provided signed, written informed consent. The subject recruitment process is depicted in Fig. 1.

### Data collection

An outpatient physician directly collected information from each expectant mother, rather than relying on patient self-reporting, to guarantee the authenticity and validity of the data. During the face-to-face interview, patients provided demographic information, age, educational attainment, occupation, parity, and household income. The height and weight of the expectant mothers were recorded during their initial antenatal visit, from which the pre-pregnancy BMI was derived. Participants were categorized into four groups based on their pre-pregnancy BMI: underweight (<18.5), normal weight (18.5–24.9), overweight (25.0–29.9), and obese (≥30.0) (*WHO Expert Consultation, 2004*). GWG was determined by calculating the difference between the initial pre-pregnancy weight and the final weight before childbirth. Following the Guidelines for Weight Monitoring and Evaluation in Pregnancy for Chinese Women (*Rasmussen et al., 2009*), expectant mothers were classified into three categories, namely, inadequate weight gain, adequate weight gain, and excessive weight gain, according to their pre-pregnancy BMI ranges.

### Nutritional assessment

We assessed the dietary quality of expectant mothers using the Chinese Healthy Dietary Index for Pregnancy (CHDI-P) (*Yang et al., 2023*). The CHDI-P scale is derived from the 2016 edition of the Dietary Guidelines for Pregnant Women in China (*Yang et al., 2018*). It was developed from evidence linking various food components with maternal health, as established through large-scale population studies and encompasses 23 food components. The validity of CHDI-P was confirmed in a cohort study involving 1,416 Chinese pregnant women (*Yang et al., 2023*). The CHDI-P scale provides a comprehensive three-dimensional evaluation of dietary quality. The first dimension, called "diversity", evaluates the consumption levels of four fundamental food groups, adhering to the core principle of dietary variety. These food groups include grains, tubers and mixed beans;
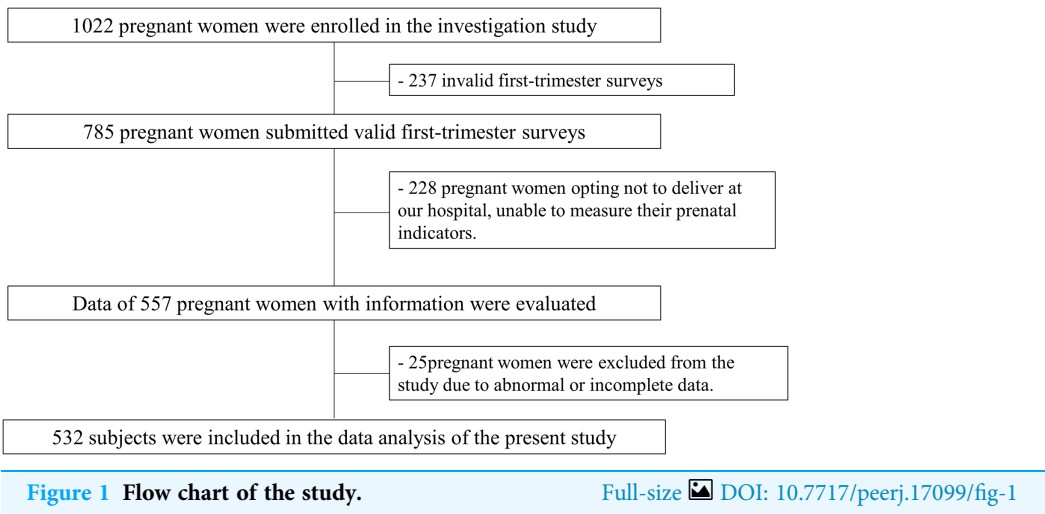

Figure 1 Flow chart of the study.               

meat, poultry, fish, and eggs; vegetables and fruits; and dairy, soybeans, and nuts. The second dimension, called "adequacy", evaluates the sufficiency of beneficial food consumption among expectant mothers. Conversely, the third dimension, called "limitation", assesses excessive consumption of proven unhealthy foods. In this study, the version employed was tailored for women in early pregnancy, featuring score ranges of 0–12 for the diversity dimension, 0–55 for the adequacy dimension, and 0–33 for the limitation dimension, culminating in an aggregate score range of 0–100. Higher scores in the diversity and adequacy dimensions correspond to an increased variety and quantity of beneficial foods consumed. Conversely, a higher score in the limitation dimension signifies reduced consumption of unhealthy foods. Therefore, lower scores are indicative of suboptimal eating habits and health.

Dietary intake was assessed using a 24-h dietary recall survey conducted over three consecutive days, encompassing 2 weekdays and 1 weekend day. In the initial survey, participants detailed the types and quantities of food consumed during face-to-face interviews with a specially trained clinician. To enhance data accuracy, participants utilized standardized measuring tools (such as bowls, spoons, and cups) and food images provided by the hospital to describe their food intake. They were also required to submit photographs of all meals consumed over the 3-day period. These photos were cross-referenced with the participants' descriptions. In cases of significant discrepancies, on-site instruction was provided to standardize the reporting process. Considering the safety of pregnant women, follow-up surveys were conducted *via* telephone. To further validate the data, participants were asked to continue providing photographs of their meals for investigator verification.

## Statistical analysis

Descriptive statistics for count data are expressed as numbers and percentages, while continuous variables are represented by means and standard deviations. Between-group comparisons for count data were conducted using the Pearson's $\chi^2$ test. For continuous variables, between-group differences were assessed using analysis of variance (ANOVA) or

the Kruskal-Wallis test, and the results were contingent upon data normality. Generalized linear models were employed to adjust for confounding variables when comparing CHDI-P scores across groups. Covariates were selected based on prior research and included age, income, educational level, smoking status, energy intake, number of pregnancies and deliveries (*Parker et al., 2019*; *Shin, Lee & Song, 2016*). *Post hoc* tests, using the Tukey-adjusted method, elucidated between-group disparities. All confidence intervals were computed at the 95% level, and a *p*-value less than 0.05 was deemed statistically significant. Statistical analyses were executed using Python 3.6.0.

# RESULTS

## Baseline characteristics

A total of 532 expectant mothers were incorporated into the study and were segmented into four categories based on their pre-pregnancy BMI: underweight (<18.5), normal weight (18.5–24.9), overweight (25.0–29.9), and obese (≥30.0). The underweight cohort were an average age of 29.29 years old (SD 4.53), and were notably younger than the other categories, with their GWG values notably surpassing those of the other groups. Significant differences were observed among the groups regarding body weight and GWG ($p < 0.001$). No statistically significant disparities were observed among the groups concerning education, family income, smoking habits, number of pregnancies or deliveries (Table 1).

The data were then categorized into three groups based on GWG: Inadequate, Adequate, and Excessive. Notably, those with adequate weight gain during pregnancy exhibited lower weights and pre-pregnancy BMI ($p < 0.001$). The remaining demographic characteristics showed no significant differences (Table 2).

## Associations between CHDI-P and pre-pregnancy BMI

Education level, household income, and number of deliveries were selected as covariates in the multivariate adjustment model. According to this model, the underweight group displayed the highest marginal mean of 60.28 ± 0.99 in CHDI-P total scores, while the obese group exhibited the lowest marginal mean of 57.80 ± 1.11 (Table 3). *Post hoc* analyses revealed significant differences in CHDI-P total scores among all groups ($p < 0.001$). Specifically, diversity total scores for both the overweight and obese groups were significantly lower than those for the normal weight group ($p < 0.001$). However, there were no significant differences in the total adequacy scores among the groups. Significant differences were observed in the limitation total scores among all groups ($p < 0.001$) (Table 4).

## Associations between CHDI-P and GWG

In the GWG model, covariates encompassed household income and the number of deliveries. The adequate and excessive group exhibited a marginal mean of 59.17 for total CHDI-P scores. Meanwhile, the marginal mean for the inadequate group stood at 59.54 ± 0.92, which surpassed the values of the other two groups (Table 5). *Post hoc* analysis of inter-group differences revealed no significant differences in CHDI-P total scores and diversity total scores among the groups. However, the total adequacy score for the

**Table 1 Basic characteristics based on pre-pregnancy BMI.**

| Characteristics | Pre-pregnancy BMI | | | | |
|---|---|---|---|---|---|
| | Underweight ($n$ = 24) | Normal weight ($n$ = 329) | Overweight ($n$ = 127) | Obese (n = 52) | $p$ |
| Age (years) | 29.29 ± 4.53 | 31.34 ± 4.18 | 32.03 ± 4.71 | 32.31 ± 4.53 | 0.017 |
| Height (cm) | 164.42 ± 5.14 | 164.19 ± 5.42 | 162.91 ± 4.85 | 163.63 ± 5.07 | 0.942 |
| Weight (kg) | 52.56 ± 9.37 | 64.65 ± 11.93 | 72.57 ± 8.74 | 81.04 ± 14.51 | <0.001 |
| GWG (kg) | 15.77 ± 5.09 | 13.93 ± 4.63 | 12.75 ± 4.20 | 12.02 ± 5.09 | <0.001 |
| Basic metabolism (kcal/d) | 1,315.41 ± 116.00 | 1,310.64 ± 150.00 | 1,360.98 ± 119.68 | 1,363.81 ± 137.76 | <0.001 |
| Educational level (N, %) | | | | | 0.382 |
| Secondary level or below | 9 (37.50) | 83 (25.23) | 46 (36.22) | 14 (26.92) | |
| College level | 7 (29.17) | 93 (28.27) | 35 (27.56) | 14 (26.92) | |
| University level | 5 (20.83) | 88 (26.75) | 32 (25.20) | 15 (28.85) | |
| Graduate level or above | 3 (12.50) | 65 (19.75) | 14 (11.02) | 9 (17.31) | |
| Household income (N, %) | | | | | 0.395 |
| <5,000 | 2 (8.33) | 16 (4.86) | 8 (6.30) | 4 (7.69) | |
| 5,000–10,000 | 3 (20.83) | 59 (17.93) | 12 (9.45) | 5 (9.62) | |
| 10,001–30,000 | 14 (58.33) | 205 (62.31) | 87 (68.50) | 38 (73.07) | |
| >30,000 | 5 (12.50) | 49 (14.89) | 20 (15.75) | 5 (9.62) | |
| Smokers (N, %) | 2 (8.33) | 35 (10.64) | 11 (8.66) | 5 (9.62) | 0.922 |
| Number of pregnancies | 2 (1, 2) | 2 (1, 2) | 2 (1, 3) | 2 (1, 3) | 0.451 |
| Number of deliveries | 1 (0, 1) | 1 (1, 1) | 1 (1, 1) | 1 (0, 1) | 0.103 |

**Table 2 Basic characteristics based on GWG.**

| Characteristics | GWG | | | |
|---|---|---|---|---|
| | Inadequate ($n$ = 25) | Adequate ($n$ = 351) | Excessive ($n$ = 156) | p |
| Age (years) | 32.28 ± 4.25 | 31.70 ± 4.67 | 31.36 ± 4.27 | 0.487 |
| Height (cm) | 163.20 ± 3.96 | 162.56 ± 5.23 | 164.46 ± 4.62 | 0.614 |
| Weight (kg) | 69.30 ± 9.80 | 62.36 ± 10.57 | 69.80 ± 13.54 | <0.001 |
| Pre-pregnancy BMI (kg/m$^2$) | 24.92 ± 3.49 | 22.86 ± 3.66 | 25.33 ± 3.69 | <0.001 |
| Basic metabolism (kcal/d) | 1,335.76 ± 97.51 | 1,334.27 ± 195.40 | 1,326.05 ± 119.20 | 0.816 |
| Educational level (N, %) | | | | 0.524 |
| Secondary level or below | 6 (24.00) | 102 (29.06) | 42 (26.92) | |
| College level | 7 (28.00) | 100 (28.49) | 44 (28.21) | |
| University level | 9 (36.00) | 83 (23.65) | 48 (30.77) | |
| Graduate level or above | 3 (12.00) | 66 (18.80) | 22 (14.10) | |
| Household income (N, %) | | | | 0.743 |
| <5,000 | 1 (4.00) | 20 (5.70) | 9 (5.77) | |
| 5,000–10,000 | 3 (12.00) | 56 (15.95) | 22 (14.10) | |
| 10,001–30,000 | 19 (76.00) | 228 (64.96) | 97 (62.18) | |
| >30,000 | 2 (8.00) | 47 (13.39) | 28 (17.95) | |

| Characteristics | GWG | | | |
|---|---|---|---|---|
| | Inadequate (*n* = 25) | Adequate (*n* = 351) | Excessive (*n* = 156) | p |
| Smokers (N, %) | 4 (16) | 34 (9.68) | 15 (9.62) | 0.587 |
| Number of pregnancies | 2 (1, 3) | 2 (1, 3) | 2 (1, 2) | 0.502 |
| Number of deliveries | 1 (1, 2) | 1 (0, 1) | 1 (1, 1) | 0.283 |

**Table 3 Multivariable adjusted CHDI-P scores by pre-pregnancy BMI.**

| | Total | Diversity total | Adequacy total | Limitation total |
|---|---|---|---|---|
| Multivariable-adjusted model—adjusted for educational level, household income and number of deliveries | | | | |
| Underweight | 60.28 ± 0.99 | 9.99 ± 0.11 | 26.26 ± 1.08 | 24.03 ± 0.32 |
| Normal weight | 59.46 ± 0.99 | 10.00 ± 0.21 | 26.26 ± 1.01 | 23.19 ± 0.38 |
| Overweight | 58.85 ± 0.98 | 9.99 ± 0.20 | 26.21 ± 1.00 | 22.64 ± 0.34 |
| Obese | 57.80 ± 1.11 | 9.99 ± 0.12 | 26.07 ± 1.15 | 21.70 ± 0.43 |

**Table 4 *Post-hoc* analysis of CHDI-P scores across BMI-based subgroups.**

| | Total | | Diversity total | | Adequacy total | | Limitation total | |
|---|---|---|---|---|---|---|---|---|
| | p | 95% CI | p | 95% CI | p | 95% CI | p | 95% CI |
| Obese-normal weight | <0.001 | [−2.04 to −1.27] | 0.004 | [−0.01 to −0.00] | 0.595 | [−0.59 to 0.21] | <0.001 | [−1.64 to −1.35] |
| Overweight-normal weight | <0.001 | [−0.87 to −0.33] | 0.007 | [−0.01 to −0.00] | 0.900 | [−0.33 to 0.22] | <0.001 | [−0.66 to −0.46] |
| Underweight-normal weight | <0.001 | [0.28–1.37] | 0.804 | [−0.01 to 0.01] | 0.900 | [−0.57 to 0.56] | <0.001 | [0.63–1.04] |
| Overweight-obese | <0.001 | [0.63–1.48] | 0.709 | [−0.00 to 0.01] | 0.836 | [−0.30 to 0.57] | <0.001 | [0.78–1.10] |
| Underweight-obese | <0.001 | [1.85–3.12] | 0.546 | [−0.00 to 0.01] | 0.876 | [−0.47 to 0.84] | <0.001 | [2.10–2.57] |
| Underweight-overweight | <0.001 | [0.85–2.00] | 0.892 | [−0.01 to 0.01] | 0.900 | [−0.54 to 0.64] | <0.001 | [1.18–1.61] |

**Table 5 Multivariable adjusted CHDI-P scores by GWG.**

| | Total | Diversity total | Adequacy total | Limitation total |
|---|---|---|---|---|
| Multivariable-adjusted model—adjusted for household income and number of deliveries | | | | |
| Inadequate | 59.54 ± 0.92 | 10.00 ± 0.11 | 26.76 ± 0.99 | 22.74 ± 0.19 |
| Adequate | 59.17 ± 0.86 | 10.00 ± 0.10 | 26.25 ± 0.98 | 22.91 ± 0.23 |
| Excessive | 59.17 ± 0.73 | 10.00 ± 0.12 | 26.19 ± 0.84 | 22.98 ± 0.24 |

inadequate group was significantly higher than that of both the adequate group (*p* = 0.023) and the excessive group (*p* = 0.006). Significant differences were observed across all groups in the limitation total scores (*p* < 0.01) (Table 6).

**Table 6 _Post-hoc_ analysis of CHDI-P scores across GWG-based subgroups.**

| | Total | | Diversity total | | Adequacy total | | Limitation total | |
|---|---|---|---|---|---|---|---|---|
| | _p_ | 95% CI | _p_ | 95% CI | _p_ | 95% CI | _p_ | 95% CI |
| Excessive-adequate | 0.900 | [−0.18 to 0.17] | 0.308 | [−0.00 to 0.01] | 0.739 | [−0.26 to 0.14] | 0.004 | [0.02–0.13] |
| Inadequate-adequate | 0.080 | [−0.03 to 0.76] | 0.521 | [−0.00 to 0.01] | 0.023 | [0.06–0.96] | 0.003 | [−0.29 to −0.05] |
| Inadequate-excessive | 0.061 | [−0.01 to 0.75] | 0.887 | [−0.00 to 0.01] | 0.006 | [0.14–1.00] | <0.001 | [−0.36 to −0.13] |

## DISCUSSION

The health of a woman during pregnancy significantly influences the lifelong health of her newborn (_Marshall et al., 2022_), and her nutritional status is of paramount importance (_Dhaded et al., 2020_). While Chinese women are becoming increasingly aware of the importance of diet during pregnancy and food scarcity is no longer a prevalent issue, the quality of their diets remains a concern due to limited access to professional guidance (_Abayomi et al., 2020_). Factors such as time constraints and inconvenient access to healthcare prevent these women from consistently monitoring and adjusting their diets. Consequently, it is crucial to identify convenient indicators for assessing dietary risks during pregnancy.

This study explored the quality of prenatal diets among a convenience sample of pregnant women in Shandong, China. Additionally, we explored the associations between pre-pregnancy BMI, GWG, and maternal dietary quality. The Alternative Healthy Eating Index for Pregnancy (AHEI-P) has been applied in previous research to quantitatively evaluate the dietary quality of pregnant women (_Hsiao et al., 2019_; _Parker et al., 2019_; _Quansah et al., 2022_). Our study sample was drawn from Shandong, China. Consequently, we utilized the CHDI-P scale, which is better tailored for Chinese pregnant women. _Parker et al. (2019)_ determined that the overall prenatal dietary quality of pregnant women was suboptimal, achieving a mean score of just 61.2 out of 130 using the AHEI-P scale. The results of our study determined that the average score for pregnant women aligned with this finding. Even after equiproportional conversion, the score indicates that the prenatal dietary quality of Chinese pregnant women remains less than ideal. Grouping by GWG did not manifest any significant differences in marginal means among the groups. Conversely, when categorized based on pre-pregnancy BMI, the underweight group had a score that exceeded the obese group by 2.5 points. While a shift of 5% in dietary quality score is necessary to deem it clinically significant (_Miller et al., 2015_), it is plausible to posit an association between pre-pregnancy BMI and dietary quality. Specifically, pregnant women with a BMI of ≥25 may be at an elevated risk for malnutrition.

In this study, the CHDI-P scale was employed to evaluate the nutritional quality of pregnant women's diets. This scale focuses on three primary dimensions: diversity, adequacy, and limitation. A diverse food intake, ensuring a broad spectrum of nutrients for both the expectant mother and the developing fetus, is fundamental to a healthy diet. Previous research has indicated that food diversity can mitigate the negative impacts of anemia and neonatal mortality linked to insufficient intake (_Lander et al., 2019_). Pregnant

women exhibiting insufficient weight gain often have less varied diets, making their nutritional habits prime targets for intervention (*Tebbani, Oulamara & Agli, 2021*). In our study, however, there were no significant differences between groups, whether categorized by pre-pregnancy BMI or by GWG. Although the *p*-value for the comparison among the obese, overweight, and normal weight groups is below 0.05, the proximity of the confidence intervals for their differences to zero renders this distinction potentially insignificant. The average scores hovered around 10 out of a possible 12. One potential explanation for this might be that this portion of the scale lacks the granularity necessary to highlight nuanced differences. Alternatively, it could suggest that pregnant women in Shandong, China, generally maintain a commendable level of dietary diversity prior to pregnancy.

The adequacy of a pregnant woman's diet is a pivotal aspect in evaluating dietary quality. The research conducted by *Cano-Ibáñez et al. (2020)* revealed that diets patterned after the Mediterranean typically exhibited greater nutrient adequacy and were associated with controlled GWG in pregnant women. Another study indicated that diets with adequate nutrients significantly curtailed metabolic complications, such as gestational diabetes (*Looman et al., 2019*). In our research, when grouping by pre-pregnancy BMI, we observed no significant disparities in adequacy scores among the groups. However, the inadequate group registered a significantly higher score compared to the other two groups. These findings align with prior research, indicating that predicting dietary adequacy in pregnant women may be more effectively gauged using GWG. In underprivileged regions, the inadequacy among pregnant women is frequently linked to insufficient intake, primarily due to limited food accessibility resulting from economic constraints (*Darling et al., 2023*). Yantai, conversely, is one of the more affluent areas in China. Given the income level of the participants in our study, we postulate that the diet of the inadequate group was likely self-regulated. These women might have enhanced the quality of their food intake while potentially emphasizing physical activity to manage their weight (*Teede et al., 2022*).

Excessive consumption of unhealthy foods adversely affects the dietary well-being of pregnant women, making restriction a crucial aspect that warrants attention. In the CHDI-P, foods that should be restricted include fried foods, sugary beverages, processed meats, alcohol, refined grains, and cooking oil. In this research, the underweight group achieved the highest scores in the limitation total, whereas the obese group recorded the lowest. A significant difference was observed across all groups. This pattern aligns with the overall trend observed in the CHDI-P total scores. While significant differences in scores were also evident when grouped by GWG, the extent of these differences was less marked compared to the pre-pregnancy BMI categorization. Previous studies provide substantial evidence on the detrimental effects of high-sugar, high-fat foods, and alcohol on pregnant women (*Pennington et al., 2020*; *Sundermann et al., 2019*; *Witek, Wydra & Filip, 2022*). Our findings indicate that pregnant women with higher BMIs tend to consume more unhealthy foods. While there is not a definitive cut-off value for CHDI-P, pregnant women in the overweight and obese categories should be made aware of the potential complications from consuming unhealthy foods compared to those with a normal BMI.

Minimizing the consumption of harmful foods could be an effective strategy to reduce complications in obese pregnant women.

Our study examined the association between pre-pregnancy BMI, GWG, and dietary quality. However, a more holistic approach might involve integrating prenatal dietary quality instead of concentrating solely on either BMI or GWG. Both pre-pregnancy BMI and GWG proved valuable in distinguishing and predicting prenatal diet quality. Notably, pre-pregnancy BMI was more sensitive to overall diet quality and restriction of certain foods, whereas GWG was more indicative of dietary adequacy. It is important to highlight that the dietary quality we assessed pertains to the early stage of a woman's pregnancy, while the GWG was determined by the difference between the initial and pre-delivery weights. The dietary habits of pregnant women typically vary throughout their pregnancy. For initial dietary quality assessment, GWG may not be as effective an indicator as BMI, especially in the middle or later stages of pregnancy. This observation warrants further investigation in subsequent research.

The distinctiveness of this study lies in its exploration of the relationship between pre-pregnancy BMI, GWG, and prenatal dietary quality within a Chinese demographic. Comparing both BMI and GWG concurrently enhances our comprehension of this association. This study has some limitations. Firstly, the sample size was limited, primarily because participants were required to deliver at our hospital. This criterion might have introduced some bias. Secondly, our data collection occurred during the COVID-19 pandemic. The impact of the pandemic on our findings was not analyzed in this study, as it would necessitate additional data exploration and further research support. Nonetheless, this was necessary to ensure uniformity across this series of studies. Another notable observation was that pregnant women accompanied by their partners were more inclined to participate and less likely to drop out.

## CONCLUSION

Our study underscores the significance of pre-pregnancy BMI and GWG as primary indicators of prenatal diet quality. Pre-pregnancy BMI provides a more sensitive measure of a pregnant woman's overall nutritional status and dietary restrictions while GWG more accurately reflects the adequacy of her food intake. We propose that integrating both measures offers a more holistic assessment of a pregnant woman's nutritional status. The findings from our study reinforce the recommendation for women to attain a normal BMI before pregnancy, as it is linked to a higher quality diet during pregnancy.

### Funding

The authors received no funding for this work.

### Competing Interests

The authors declare that they have no competing interests.

## Author Contributions

- Juan Zhang analyzed the data, prepared figures and/or tables, authored or reviewed drafts of the article, and approved the final draft.
- Xue Wang analyzed the data, prepared figures and/or tables, and approved the final draft.
- Ping Zhu performed the experiments, analyzed the data, prepared figures and/or tables, and approved the final draft.
- Xiaoge Huang performed the experiments, analyzed the data, prepared figures and/or tables, and approved the final draft.
- Xingru Cao conceived and designed the experiments, authored or reviewed drafts of the article, and approved the final draft.
- Junmin Li conceived and designed the experiments, authored or reviewed drafts of the article, and approved the final draft.

## Human Ethics

The following information was supplied relating to ethical approvals (*i.e.*, approving body and any reference numbers):

Jinan Maternal and Child Ethics Committee Health Hospital.

## Data Availability

The raw data is available in the Supplemental File.

## Supplemental Information

Supplemental information for this article can be found online at http://dx.doi.org/10.7717/peerj.17099#supplemental-information.

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
