# Peer review of "Exploring the relationships between pre-pregnancy BMI, gestational weight gain, and nutritional intake: a real-world investigation in Shandong, China"

_PeerJ, doi:10.7717/peerj.17099_

## Round 0.1 · original submission · Major Revisions

Please revise the manuscript as the reviewers suggested.

**Language Note:** The review process has identified that the English language must be improved. PeerJ can provide language editing services - please contact us at copyediting@peerj.com for pricing (be sure to provide your manuscript number and title). Alternatively, you should make your own arrangements to improve the language quality and provide details in your response letter. – PeerJ Staff

Reviewer 1 ·

Basic reporting

In this study, the authors provide insights into the association between prenatal diet quality and pre-pregnancy BMI and GWG based on pregnant women in Shandong, China. The topic is important and the study is generally well-designed. However, this study should be carefully revised before further consideration. My comments are given below.

Experimental design

This study focused on the two indicators of BMI and GWG. However, the final conclusion does not provide a good summary of the advantages and disadvantages of these two indicators. The authors should improve their conclusion to illustrate further the results after comparison.
The recruitment process should be standardized, and a graphical representation is necessary.The utilization of the CHDI-P scale needs further explanation to affirm its validity in reflecting the dietary status of pregnant women in China.

Validity of the findings

The topic is important and the study is generally well-designed.

Additional comments

Major comments
(1) This study focused on the two indicators of BMI and GWG. However, the final conclusion does not provide a good summary of the advantages and disadvantages of these two indicators. The authors should improve their conclusion to illustrate further the results after comparison.
(2) The recruitment process should be standardized, and a graphical representation is necessary.
(3) The utilization of the CHDI-P scale needs further explanation to affirm its validity in reflecting the dietary status of pregnant women in China.
(4) The dietary habits and nutritional needs of pregnant women vary from one pregnancy to another. Have the authors considered this issue properly? Even though the criteria for subject enrollment is labelled up to 12 weeks.


Minor revisions are needed:
(5) Language enhancements are strongly advised to facilitate reader comprehension.
(6) The sample size of 532 patients is really not a large enough number of patients to justify the authors' conclusions based on this number of patients?
(7) Why was the inclusion criterion ‘plan to give birth at the hospital’ necessary?
(8) Other factors, such as ethnicity, should be considered in the collection of basic information, and the time of collection was during COVID-19, which was not mentioned in this study, so the discussion section should have been more comprehensive.

Reviewer 2 ·

Basic reporting

This study presents a pioneering and practical examination of the relationship between gestational weight gain (GWG), pre-pregnancy Body Mass Index (BMI), and prenatal dietary quality among 532 pregnant women in Shandong Province, China. The use of GWG and BMI, which are readily accessible and standard measurements from expectant mothers, is a smart choice. These factors have been suspected to be linked with dietary habits, but the nature of this relationship has remained somewhat unclear. Your thorough analysis sheds valuable light on how GWG and BMI can predict prenatal dietary quality, offering important insights for designing interventions aimed at improving nutrition among overweight and obese pregnant women.

Experimental design

The methodological thoroughness of your paper is commendable. The process of participant recruitment could be described in more detail. It would be beneficial to include information about the initial number of participants, the attrition rates, and explanations for these figures to enhance the transparency and reliability of your study.

A more detailed description of your data collection methods would be valuable. Clarifying the steps involved in dietary recall and detailing the training protocols for interviewers would help reinforce the authenticity and methodological robustness of your data.

Validity of the findings

Expanding the discussion section would significantly strengthen your paper. By incorporating more recent studies from the last three to five years, you can provide a richer context and make a more substantial academic contribution.

Additional comments

Improving the English language usage, particularly in terms of grammatical collocations, will greatly benefit the paper, ensuring it aligns more closely with the standards expected in scholarly publications.

Reviewer 3 ·

Basic reporting

Thank you for the opportunity to review the study "Exploring the relationships between pre-pregnancy BMI, gestational weight gain, and nutritional intake: a real-world investigation in Shandong, China". In my opinion, the paper is a good piece of work, well structured, clear and easy to follow and understand. The introduction section is well presented and provides a contextualization of the topic addressed. The raw data was shared in the supplemental materials section. The results from the study are clearly presented, with professional looking tables. The discussion is structured, brings relevant comparisons with previous studies and includes limitations. Most references used are current and relevant.

Experimental design

The research question is well defined and relevant.
According to the authors, this study received approval from the Jinan Maternal and Child Ethics Committee Health Hospital (n. 2023-1-029) and all participating individuals furnished their signed, written informed consent.
Overall, the methods and results are detailed and well written; however I may have some comments/suggestions about this topic:
(1) Why age was not included as a covariate in the multivariate model that assessed the association between CHDI-P and pre-pregancy BMI? Since differences were found in table 1?
(2) Regarding this, how were the covariates that remained in the final model selected?

Validity of the findings

The findings seem to be valid. Conclusions are well stated and are linked to the research question.

Additional comments

Minor comments/suggestions:
(1) I suggest organizing the table numbers following the order they appear in the text.
(2) The title of all tables is described as “Table 1”. Is it correct?

---

## Round 0.2 · accepted · Accept

This manuscript can be accepted now.

Reviewer 1 ·

Basic reporting

In this study, the authors analyzed a sample of 532 women registered at an outpatient clinic who were in the early stages of pregnancy. The results revealed that the underweight group had significantly higher total scores and limitation total scores on the CHDI-P (P<0.001). Conversely, the overweight and obese groups were more susceptible to suboptimal dietary quality. Notably, the inadequate weight gain group displayed significantly elevated food adequacy scores compared to the other two groups (p<0.05). These results indicate that greater GWGs do not necessarily align with the principles of adequate nutrition. The authors have revised this paper according to my suggestions.

Experimental design

no comment

Validity of the findings

no comment

Reviewer 2 ·

Basic reporting

no comment

Experimental design

no comment

Validity of the findings

no comment

Additional comments

I have no further recommendation. The article has been greatly improved and is worthy of publication.